# Structural and functional characterization of an otopetrin family proton channel

Qingfeng Chen[1,2,3], Weizhong Zeng[1,2,3], Ji She[1,2], Xiao-chen Bai[2,4]*, Youxing Jiang[1,2,3]*

[1]Department of Physiology, University of Texas Southwestern Medical Center, Dallas, United States; [2]Department of Biophysics, University of Texas Southwestern Medical Center, Dallas, United States; [3]Howard Hughes Medical Institute, University of Texas Southwestern Medical Center, Dallas, United States; [4]Department of Cell Biology, University of Texas Southwestern Medical Center, Dallas, United States

**Abstract** The otopetrin (OTOP) proteins were recently characterized as proton channels. Here we present the cryo-EM structure of OTOP3 from *Xenopus tropicalis* (XtOTOP3) along with functional characterization of the channel. XtOTOP3 forms a homodimer with each subunit containing 12 transmembrane helices that can be divided into two structurally homologous halves; each half assembles as an $\alpha$-helical barrel that could potentially serve as a proton conduction pore. Both pores open from the extracellular half before becoming occluded at a central constriction point consisting of three highly conserved residues – $Gln_{232/585}$-$Asp_{262}$/$Asn_{623}$-$Tyr_{322/666}$ (the constriction triads). Mutagenesis shows that the constriction triad from the second pore is less amenable to perturbation than that of the first pore, suggesting an unequal contribution between the two pores to proton transport. We also identified several key residues at the interface between the two pores that are functionally important, particularly Asp509, which confers intracellular pH-dependent desensitization to OTOP channels.
DOI: https://doi.org/10.7554/eLife.46710.001

*For correspondence:
Xiaochen.Bai@UTSouthwestern.edu (X-B);
youxing.jiang@utsouthwestern.edu (YJ)

**Competing interests:** The authors declare that no competing interests exist.

## Introduction

Proton transport plays an important role in a number of biological processes, including ATP synthesis (*Walker, 2013*), host defense (*Morgan et al., 2009*), sperm mobility (*Lishko et al., 2010*), viral entry (*Bron et al., 1993*) and sour taste perception (*Tu et al., 2018*). So far, only a handful of proton channels have been identified (*Decoursey, 2003*): the influenza A M2 channel, which is required for viral entry and replication (*Bron et al., 1993*; *Pinto et al., 1992*), and the animal hv1 channel, which is involved in host defense through acidification of the phagosome that eventually inactivates the infectious agents (*Morgan et al., 2009*; *Ramsey et al., 2006*; *Sasaki et al., 2006*).

The perception of taste is mediated by receptors that are expressed on the surface of taste cells in the tongue (*Liman et al., 2014*; *Yarmolinsky et al., 2009*). While the taste receptors for sensing sweet, bitter, umami and low salt have been identified (*Zhang et al., 2003*), the identity of the sour receptor has been controversial. The PKD2L1 channel, which shows specific expression in type three taste cells where sour taste is perceived, was initially proposed as the sour receptor (*Huang et al., 2006*) but was later refuted based on the findings that PKD2L1 knock-out mice retained most of their sensitivity to sour taste perception (*Horio et al., 2011*). A recently characterized proton channel called otopetrin1 (OTOP1) appears to be the bona fide sour receptor because it expresses specifically in type three taste cells rather than other taste cell types, generates a proton current across the plasma membrane in response to extracellular acidification, and exhibits the $Zn^{2+}$ sensitivity that is a key feature observed in the proton current of sour taste cells (*Tu et al., 2018*; *Chang et al., 2010*; *Bushman et al., 2015*; *Ramsey and DeSimone, 2018*; *Montell, 2018*).

OTOP1 was initially identified for its role in the development of otoconia, which are calcium carbonate-based structures that sense gravity and acceleration in the vestibular system (*Hughes et al., 2006*), likely by regulating cellular calcium homeostasis (*Hughes et al., 2007*; *Kim et al., 2010*). Three mutations in mouse OTOP1 – *tilted (tlt)* (*Ornitz et al., 1998*), *mergulhador (mlh)* (*Hurle et al., 2003*), and *inner ear defect (ied)* (*Besson et al., 2005*) – caused vestibular defects in mouse. OTOP1 was also found to play a role in otolith formation in zebrafish through the characterization of the *backstroke (bks)* mutation that causes a complete absence of otoliths (*Hughes et al., 2004*; *Söllner et al., 2004*). Additionally, OTOP2 and OTOP3 represent two main paralogs of OTOP1 in mouse and other mammals (*Hurle et al., 2011*). While the tissue-specific patterns of expression of OTOP2 and OTOP3 are distinct from OTOP1, their cellular functions remain elusive (*Tu et al., 2018*; *Hurle et al., 2011*). In this study, we determined the cryo-EM structure of XtOTOP3, which adopts a unique double barrel fold, thus, defining the overall architecture of the otopetrin family of ion channels. Additionally, our mutagenesis and electrophysiological analysis allowed us to pinpoint several important regions within the channel that define the unique biophysical properties of the OTOP family of proton channels.

## Results

### Functional characterization of XtOTOP3

N-terminal GFP-tagged XtOTOP3 was transiently expressed in HEK293 cells for electrophysiology studies. The majority of the expressed proteins were localized to the plasma membrane as demonstrated by confocal imaging (*Figure 1a*). As shown in whole-cell recordings at −100 mV, lowering the extracellular pH elicited inward proton currents conducted by XtOTOP3, with significantly larger currents at a pH below 5.0, indicating extracellular pH-dependent channel activation (*Figure 1b*). The reversal potentials measured from the I-V curves at various extracellular pH values match closely with the calculated Nernst potentials of the pH gradient across the plasma membrane (*Figure 1c*). Furthermore, the reversal potential was not affected by the replacement of 150 mM extracellular $NMDG^+$ with $Li^+$, $K^+$ or $Cs^+$ (*Figure 1—figure supplement 1*), confirming the proton selectivity of XtOTOP3.

Similar to that observed in mouse OTOP1 (MmOTOP1), the proton currents of XtOTOP3 decayed rapidly, and larger inward proton current resulted in a faster decay. Two experimental observations led us to conclude that this current decay is caused by both rapid intracellular local proton accumulation upon channel opening and intracellular pH-dependent desensitization of the channel. In one experiment, we measured channel activity with a constant extracellular pH ($pH_o$) of 4.5 but with varying intracellular pH ($pH_i$). As shown in *Figure 1d*, the channel is barely conductive when the intracellular pH is maintained at 6.0 but elicited a large proton current when the intracellular pH is adjusted to 6.5 or higher, indicating that XtOTOP3 becomes inactive when intracellular pH becomes acidic. In a second experiment, the intracellular (pipette) pH is maintained at 7.4 and the extracellular (bath) pH is set to 5.0, and the proton currents were elicited by repetitive voltage ramps from −100 mV to 100 mV over a 150 ms duration with a 90 ms interval between two pulses. As shown in the resulting I-V curves, the reversal potential is shifted progressively towards 0 mV, indicating a decrease in the proton gradient that results from the accumulation of localized intracellular protons (*Figure 1e*). Meanwhile, the proton currents become progressively smaller because of the channel desensitization at lower intracellular pH caused by a local proton accumulation (*Figure 1e*). This pH-dependent channel desensitization also explains the faster current decay associated with the larger inward proton currents (*Figure 1b*).

$Zn^{2+}$, but not $Ca^{2+}$, has been shown to partially block MmOTOP1 from the extracellular side. We, therefore, also tested the effects of divalent blockage on XtOTOP3. Distinct from MmOTOP1, our results demonstrated that both $Zn^{2+}$ and $Ca^{2+}$ can partially block the channel (*Figure 1f,g*). Given the high extracellular $Ca^{2+}$ concentration under physiological conditions, the $Ca^{2+}$ blockage would modulate the conductance of the channel and could potentially be a physiologically relevant property of XtOTOP3.

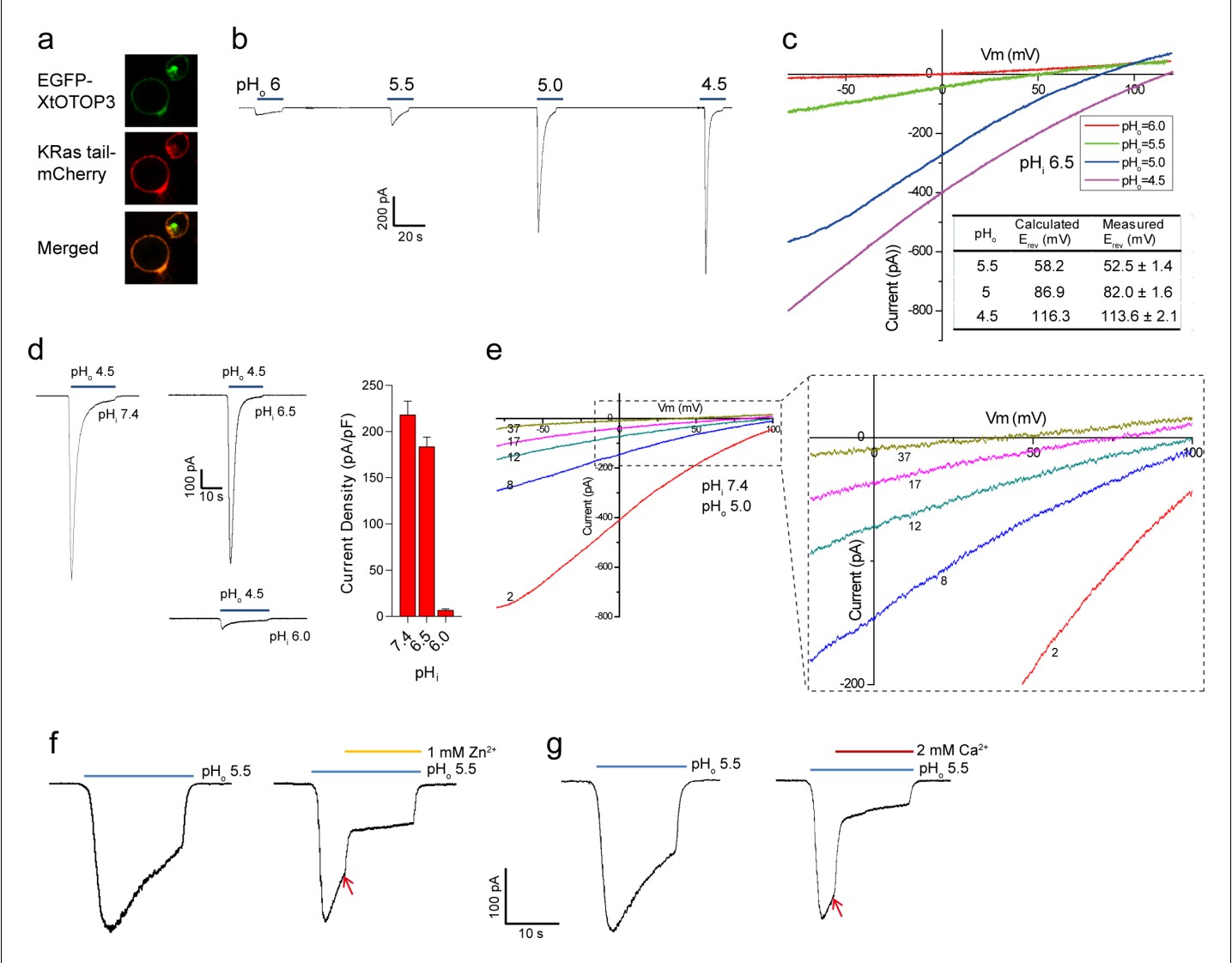

**Figure 1.** Functional characterization of XtOTOP3. (**a**) Plasma membrane localization of XtOTOP3 expressed in HEK293 cells. (**b**) Sample traces of whole-cell currents measured at $-100$ mV in HEK-293 cell expressing XtOTOP3 in Na$^+$-free extracellular solution. Currents were elicited by changing the extracellular pH (pH$_o$) from 7.4 to various acidic pH. Intracellular pH (pH$_i$) was set at 7.4. (**c**) Sample I-V curves of XtOTOP3 with varying extracellular pH. The intracellular pH is set at 6.5. Inset shows the calculated and measured reversal potentials at pH$_o$ = 4.5–5.5. (**d**) Sample traces of the whole-cell currents measured at $-100$ mV with pH$_o$ = 4.5 and pH$_i$ = 6.0, 6.5 and 7.4 (left), and the current density at various pH$_i$ (right). (**e**) Sample I-V curves obtained by repetitive voltage ramps from $-100$ mV to 100 mV over a 150 ms duration with 90 ms intervals at pH$_i$ = 7.4 and pH$_o$ = 5.0. We used short ramp duration for I-V curves to reduce the total proton influx during each ramp and thereby slow down the current decay, allowing us to observe the progressive change in channel currents and the reversal potentials. Only representative I-V curves indicated by the voltage ramp numbers are shown. The expanded view (inset) highlights the progressive decrease in reversal potentials. (**f**) Sample traces of whole-cell currents recorded in the absence (left) and presence (right) of 1 mM extracellular Zn$^{2+}$. pH$_i$ is set to 7.4 and the currents were elicited by changing pH$_o$ to 5.5 to reduce the rate of desensitization. (**g**) Sample traces of whole-cell currents recorded in the absence (left) and presence (right) of 2 mM extracellular Ca$^{2+}$. Arrows indicate the time when divalent cations were introduced. Data in c and d are shown as mean ± s.e.m. (n = 5 independent experiments). All recordings shown here were performed using patch clamp in the whole-cell configuration.

DOI: https://doi.org/10.7554/eLife.46710.002

The following source data and figure supplement are available for figure 1:

**Source data 1.** Source data for *Figure 1d*.
DOI: https://doi.org/10.7554/eLife.46710.004
**Figure supplement 1.** Proton selectivity of XtOTOP3.
DOI: https://doi.org/10.7554/eLife.46710.003

## Structure determination of XtOTOP3

To gain further insight into the OTOP family of ion channels, we purified XtOTOP3 and determined its structure to 3.9 Å resolution using single-particle cryo-EM (*Figure 2a*, *Figure 2—figure supplement 1*, *Figure 2—source data 1*). The overall density map was of sufficient quality for de-novo model building for major parts of the protein (*Figure 2—figure supplement 2*). XtOTOP3 forms a dimer, with each subunit containing 12 transmembrane helices (TMs) (*Figure 2b–d*, *Figure 2—figure supplement 3*). The N- and C- termini of OTOP channels have been predicted to face the cytosolic side (*Hughes et al., 2008*). We confirmed the orientation of the heterologously expressed XtOTOP3 in HEK293 cells using a fluorescence protease protection (FPP) assay. It was shown in the assay that plasma membrane-targeted XtOTOP3 containing an N-terminal GFP, initially resistant to

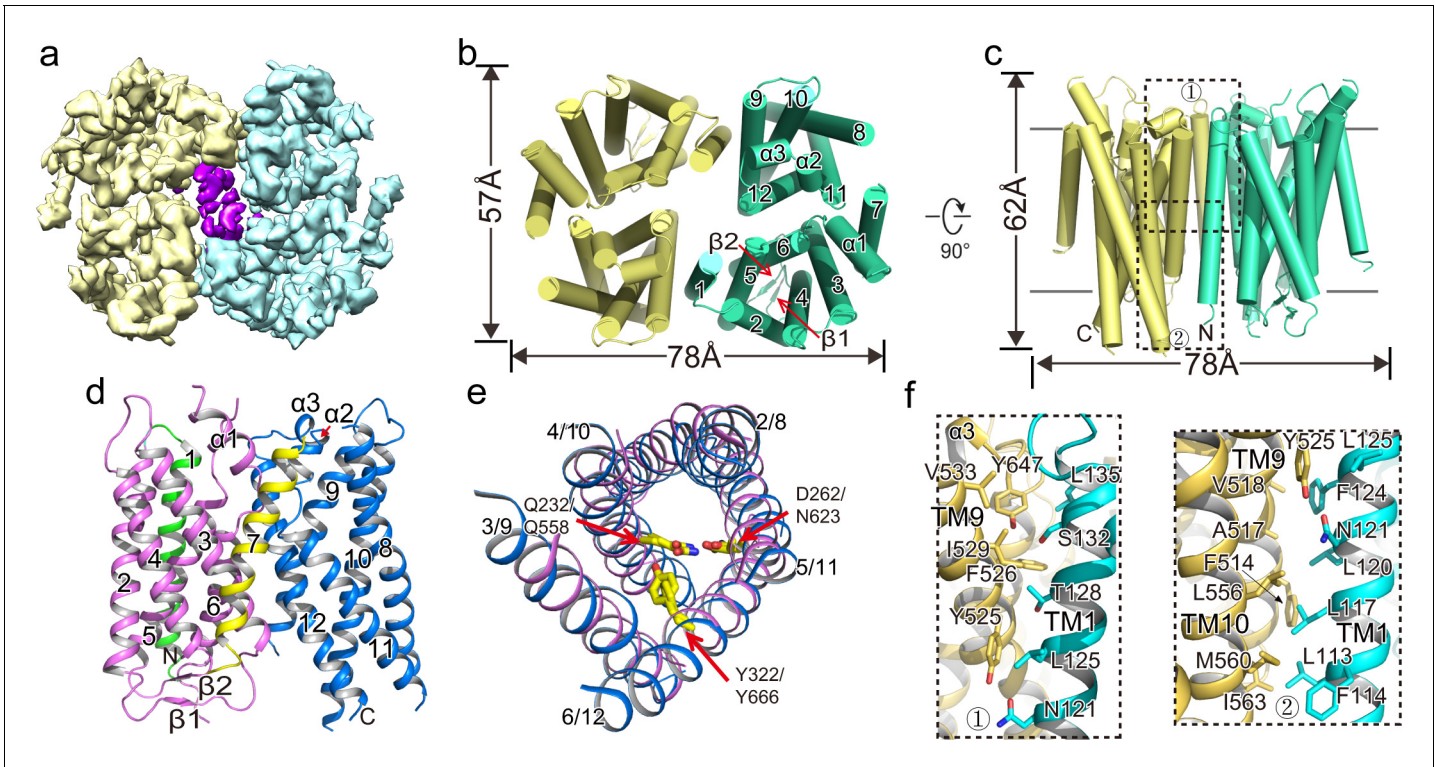

**Figure 2.** Overall structure of XtOTOP3. (a) Top view of a 3D reconstruction of XtOTOP3 with each subunit individually colored and lipid density shown in purple. (b) Top view of a cylinder representation of the XtOTOP3 structure, with the secondary structural elements labeled in one subunit. (c) Side view of a cylinder representation of the XtOTOP3 structure. (d) Cartoon representation of a single subunit of the XtOTOP3 structure with secondary structural elements labeled. The two α-barrels from the N- and C-halves are colored purple and blue, respectively. (e) Superimposition of the two α-barrels within each subunit, with the TM numbers labeled and the two sets of constriction triads shown as yellow sticks. Loops are removed for the superimposition. (f) Zoomed-in view of the inter-subunit dimerization contacts at the two regions marked by the dotted boxes in c.
DOI: https://doi.org/10.7554/eLife.46710.005

The following source data and figure supplements are available for figure 2:

**Source data 1.** Cryo-EM data collection, refinement and validation statistics.
DOI: https://doi.org/10.7554/eLife.46710.010

**Figure supplement 1.** Structure determination of XtOTOP3.
DOI: https://doi.org/10.7554/eLife.46710.006

**Figure supplement 2.** Sample EM density maps (blue mesh) for various parts of the channel.
DOI: https://doi.org/10.7554/eLife.46710.007

**Figure supplement 3.** Multiple sequence alignment of otopetrin proton channels.
DOI: https://doi.org/10.7554/eLife.46710.008

**Figure supplement 4.** Fluorescence Protease Protection (FPP) assay.
DOI: https://doi.org/10.7554/eLife.46710.009

extracellular protease cleavage, was proteolyzed following cell permeabilization by digitonin, confirming the N-terminus inside orientation (*Figure 2—figure supplement 4*).

Each XtOTOP3 subunit can be divided into two structurally homologous halves, each containing 6 TMs. The two halves of the channel assemble into a double-barrel architecture in which the latter 5 TMs from each half (TMs 2–6 in the N-half and TMs 8–12 in the C-half) form individual α-helical barrels and their respective first TMs (TM1 and TM7) are positioned on the periphery (*Figure 2b–d*). Despite their low sequence similarity, the two side-by-side α-helical barrels within each subunit share a similar structure (*Figure 2e*). Furthermore, a structure homology search using DALI (*Holm and Rosenström, 2010*) did not identify any protein with a similar structural arrangement, indicating that the structure of XtOTOP3 represents a new fold.

The dimerization of XtOTOP3 is mediated by extensive hydrophobic contacts between TM1 of one subunit and TMs 9–10 from a neighboring subunit, with a calculated surface contact area of 1536 $Å^2$ (*Figure 2f*). The XtOTOP3 dimer encloses a central cavity that is highly hydrophobic and is likely filled with lipids; indeed, elongated lipid-like densities were observed within the cavity (*Figure 2a*). However, the functional relevance of the channel's dimerization remains to be tested.

## The putative proton conduction pores

The two membrane-spanning barrels within each XtOTOP3 subunit form two potential ion conduction pores – named the N-pore and C-pore – that appear to be structurally independent from one another. From the extracellular half of the plasma membrane, both pores contain solvent accessible cavities that are enclosed by TMs 2–6 (N-pore) and TMs 8–12 (C-pore), respectively (*Figure 3a*). On the intracellular half, the pores become occluded due to the formation of two tightly packed 4-helical bundles by TMs 2, 4–6 in the N-pore and TMs 8, 10–12 in the C-pore (*Figure 3a*). The extracellular opening of the C-pore is partially covered by a linker containing two short helices (α2 and α3) that connects TM11 and TM12 (*Figure 2b and 3a*).

Moving down from the extracellular to intracellular side, the two pores within each subunit transition from the open to occluded state midway in the membrane, with constriction points formed by three conserved residues named the constriction triads: Gln232/Asp262/Tyr322 in the N-pore and Gln585/Asn623/Tyr666 in the C-pore (*Figure 3b–c*). Asparagine is observed at the Asp262 position in most OTOP channels (*Figure 2—figure supplement 3*) and an Asp262Asn mutation in XtOTOP3 exhibits similar channel function as that of the wild type (*Figure 3d*). The two sets of constriction triads are equivalently positioned in each pore, as demonstrated in the superimposition of the two pores (*Figure 2e*).

As neither of the pores provides a conductive pathway, we performed mutagenesis on the two conserved constriction triads to probe how the two pores contribute to proton conduction. While most of the mutations tested have a profound effect on channel activity, the N-pore constriction triad appears to be more amenable to mutations than that of the C-pore, as most of the mutant channels still conducted protons (*Figure 3d*). On the contrary, the C-pore constriction triad exhibits greater sensitivity to mutagenesis; in particular, most mutations to Gln585 and Asn623 yielded virtually no currents (*Figure 3e*). Two distinct sequence and structural features in the vicinity of the C-pore constriction triad are worth noting. First, the Tyr666 of the C-pore constriction triad comes from the highly conserved FYR motif (*Hughes et al., 2008*) of the OTOP channels, which is present in the C-pore but not the N-pore (*Figure 3b–c* & *Figure 2—figure supplement 3*). Second, the C-pore constriction triad forms an extensive hydrogen-bonding network with some conserved surrounding residues that are not present in the N-pore (*Figure 3b–c*). These surrounding residues include Gln520, Gln554 and Arg667, among which Arg667 is from the FYR motif and Gln554 is at the equivalent position of the *backstroke* mutant (Glu429Val) in zebrafish OTOP1 (*25, 26*) (*Figure 3c*). Collectively, our structural and functional analysis of the two constriction triads suggests that, although both pores could potentially function as proton permeation pathways, the triad in the C-pore appears to play a more determinant, if not exclusive, role in the channel activity of XtOTOP3.

## Interface between the N-pore and C-pore

Interestingly, the interface between the two pores is only tightly packed on the extracellular half of the channel but is solvent accessible from the intracellular half (*Figure 4a*). Several highly conserved

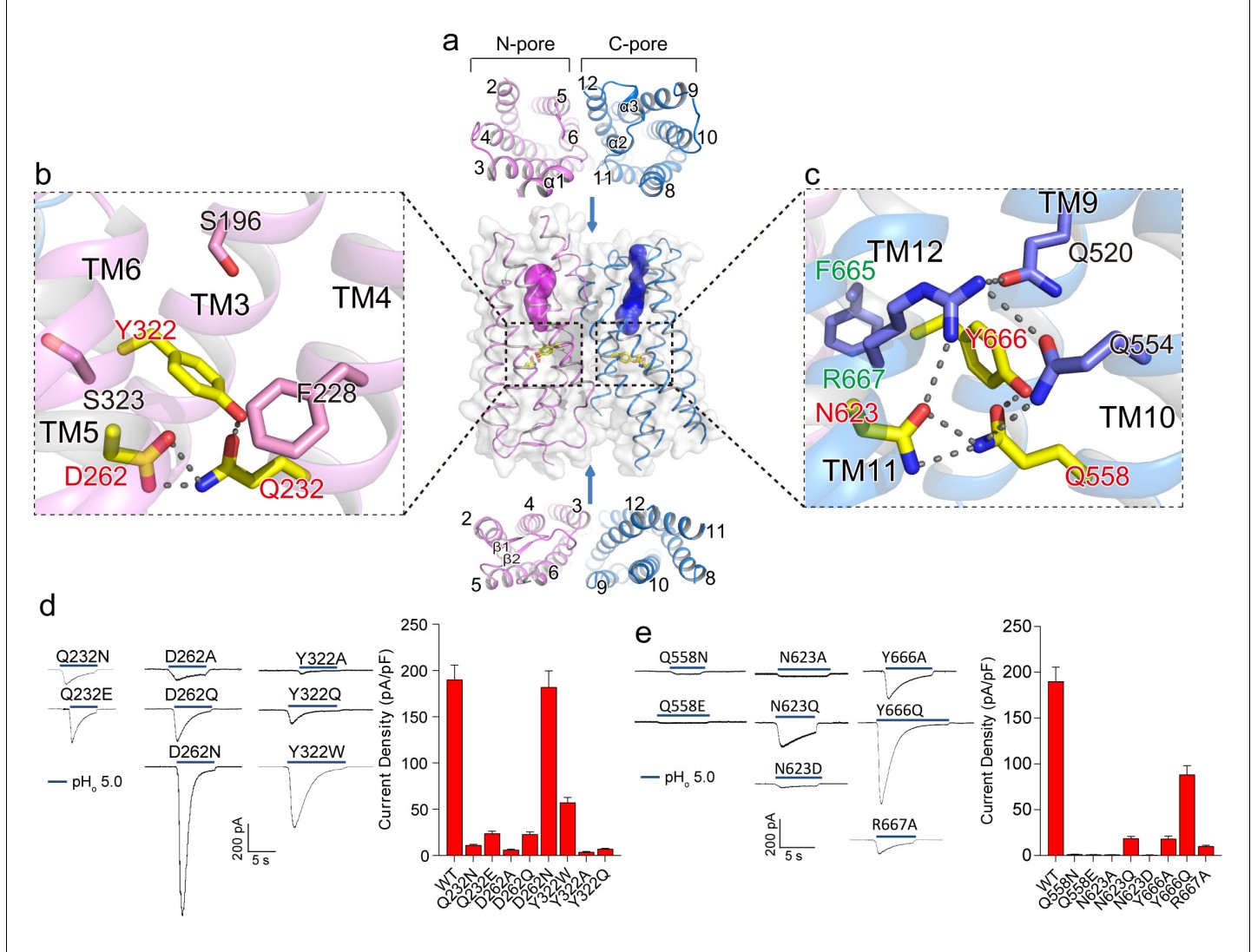

**Figure 3.** Putative proton conduction pores. (**a**) The surface-rendered XtOTOP3 structure shows two solvent accessible openings (magenta and blue) from the extracellular side. Upper and lower insets show membrane-spanning barrels viewed from the extracellular and intracellular side, respectively. (**b**) Zoomed-in view of the Gln-Asp-Tyr constriction triad (yellow) and surrounding residues (pink) in the N-pore. Hydrogen-bonds are shown as dashed lines. (**c**) Zoomed-in view of the Gln-Asn-Tyr constriction triad (yellow) and surrounding residues (blue) in the C-pore. Dashed lines mark the hydrogen-bonding network. (**d**) Sample traces of whole-cell currents of the N-pore triad mutants measured at −100 mV (left) and their current densities (right). The currents were elicited by changing $pH_o$ from 7.4 to 5.0 with $pH_i$ = 7.4. (**e**) Sample traces of the C-pore triad mutants (left) and their current densities (right) using the same recording conditions as d. Data in d and e are shown as mean ± s.e.m. (n = 5 independent experiments).

DOI: https://doi.org/10.7554/eLife.46710.011

The following source data is available for figure 3:

**Source data 1.** Source data for *Figure 3d and e*.
DOI: https://doi.org/10.7554/eLife.46710.012

residues among OTOP channels are located at this interface (*Hughes et al., 2008*) (*Figure 2—figure supplement 3*). Consisting of predominantly hydrophobic residues from TMs 3 and 6 of the N-pore and TMs 11 and 12 of the C-pore, the interface on the extracellular half maintains the tight packing between the two pores (*Figure 4b*). Intriguingly, a conserved acidic residue, Glu321 (*Figure 2—figure supplement 3*), is positioned at the center of this highly hydrophobic interface (*Figure 4b*). Intuitively, we expect a buried acidic residue to destabilize the inter-pore interactions; however, our mutagenesis study showed that Glu321 can be replaced with Asp without affecting channel activity while its neutralization (Glu321Gln) completely abolished channel activity (*Figure 4d*). In light of the

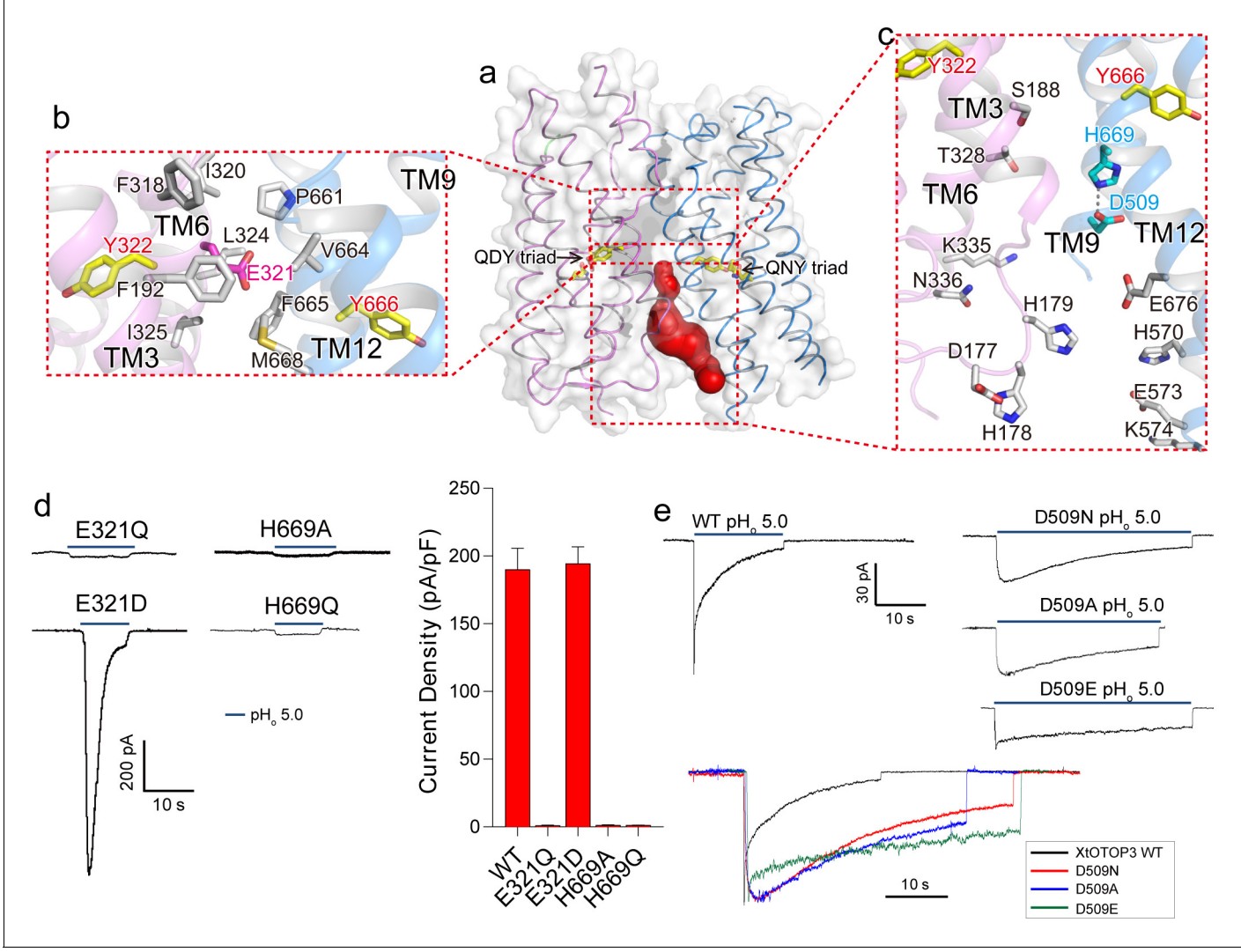

**Figure 4.** Inter-pore interface. (**a**) The surface-rendered XtOTOP3 structure shows a solvent accessible cavity (red) from the intracellular side at the interface between the two pores. The Gln-Asp/Asn-Tyr constriction triads are shown as yellow sticks. (**b**) Zoomed-in view of the inter-pore protein contacts on the extracellular half of the membrane, highlighting the conserved glutamate (Glu321, magenta sticks) buried within an interface formed mostly by hydrophobic residues (grey sticks). (**c**) Zoomed-in view of the surrounding residues of the intracellular cavity. The two hydrogen-bonded residues, Glu509 and His669, are shown as cyan sticks. (**d**) Sample traces of whole cell recordings (left) of mutations at Glu321 and His669, and their current densities (right). (**e**) The Asp509 neutralization mutants exhibit reduced desensitization rate. Top, Sample traces of patch clamp recordings in the outside-out configuration of the wild type XtOTOP3 and the Asp509 mutants. Bottom, Overlaid sample traces with mutant traces scaled to that of the wild-type. Data in d are shown as mean ± s.e.m. (n = 5 independent experiments). All recordings were measured at −100 mV with $pH_i$ = 7.4. The currents were elicited by changing $pH_o$ from 7.4 to 5.0.

DOI: https://doi.org/10.7554/eLife.46710.013

The following source data and figure supplements are available for figure 4:

**Source data 1.** Source data for *Figure 4d*.

DOI: https://doi.org/10.7554/eLife.46710.016

**Figure supplement 1.** The Asp509 mutations of XtOTOP3 reduce the rate of desensitization.

DOI: https://doi.org/10.7554/eLife.46710.014

**Figure supplement 2.** The Asp388 neutralization mutation of MmOTOP1 also reduces the channel's rate of desensitization.

DOI: https://doi.org/10.7554/eLife.46710.015

mutagenesis results, along with its high conservation, it appears that the intriguing localization of Glu321 to the hydrophobic inter-pore interface is essential for OTOP function.

Containing multiple charged and hydrophilic residues from TMs 3 and 6 of the N-pore and TMs 9 and 12 of the C-pore, the intracellular half of the inter-pore interface does not mediate any tight packing between the two pores but, instead, encloses an intracellular solvent accessible cavity (*Figure 4c*). Two highly conserved interfacial residues, D509 on TM9 and His669 on TM12, form a hydrogen-bond and are positioned at the bottom of the cavity with close proximity to the C-pore constriction triad (*Figure 4c*). Mutations to His669 (His669Ala and His669Gln) completely abolished channel activity (*Figure 4d*). Interestingly, Asp509Asn and Asp509Ala mutations both resulted in a channel that exhibits much slower desensitization (*Figure 4e* and *Figure 4—figure supplement 1*). The slower rate of desensitization exhibited by these Asp509 mutations is not merely caused by the loss of a negatively charged, titratable carboxylate group, as the Asp509Glu mutation also exhibits a similar slow desensitization as the neutralization mutation. A similar neutralization mutation to the equivalent residue in mouse OTOP1 (Asp388Asn) also reduced the rate of channel desensitization (*Figure 4—figure supplement 2*), suggesting a general role for this acidic residue in the highly characteristic, intracellular pH-dependent desensitization among OTOP family members.

## Discussion

By solving the cryo-EM structure of XtOTOP3, we define the overall architecture of the recently characterized OTOP proton channel family (*Tu et al., 2018*). The structure adopts a unique two-pore architecture in which both pores could potentially provide pathways for proton permeation. Given the observed constrictions in both pores of XtOTOP3, we suspect that our current structure likely represents the closed state. However, it is worth noting that our current structural and mutagenesis analysis have not yet provided direct evidence to prove that one or both of the pores indeed serve as the ion conduction pathway. It is also unclear whether the triads in both pores are directly involved in proton conduction. Furthermore, given the observation that mutations at either triad can have a profound effect on channel activity, we could not rule out an alternative possibility that the permeation pathway is completely different from the two pores observed in the structure. One possibility is that the mutations to the two constriction triads may allosterically modulate a single, yet unidentified pathway for proton permeation. Thus, what a conductive channel looks like and how protons are selectively relayed from the extracellular to intracellular side require further investigation.

A defining characteristic among OTOP family members is their pH-dependent activation and rapid desensitization (*Tu et al., 2018*). In this study, we show that pH regulates the activity of XtOTOP3 from both sides of the plasma membrane. Low pH activates the channel from the extracellular side but inactivates the channel on the intracellular side. Our functional analysis of XtOTOP3 led us to identify several key regions within the channel that are crucial to channel activity – particularly Asp509, which markedly reduced the rate of desensitization among OTOP family members. Thus, the structure of XtOTOP3, along with the functional hotspots that we have identified in our study, will serve as a valuable template for further mechanistic studies into the OTOP family of proton channels.

## Materials and methods

**Key resources table**

| Reagent type (species) or resource | Designation | Source or reference | Identifiers | Additional information |
|---|---|---|---|---|
| Gene (*Xenopus Tropicalis*) | OTOP3 | Tranomic technologies | NCBI: NP_0011 06584.1 | |
| Cell line (*Homo sapiens*) | Freestyle 293 F | Thermo Fisher Scientific | RRID: CVCL_D603 | |

*Continued on next page*

*Continued*

| Reagent type (species) or resource | Designation | Source or reference | Identifiers | Additional information |
|---|---|---|---|---|
| Cell line (*Spodoptera frugiperda*) | Sf9 | Thermo Fisher Scientific | RRID: CVCL_0549 | |
| Recombinant DNA reagent | pEZT-BM | DOI: 10.1016/j.str.2016.03.004 | Addgene: 74099 | |
| Software, algorithm | MotionCor2 | DOI: 10.1038/nmeth.4193 | | |
| Software, algorithm | GCTF | DOI: 10.1016/j.jsb.2015.11.003 | | |
| Software, algorithm | RELION2 | DOI: 10.7554/eLife.18722 | | |
| Software, algorithm | Coot | DOI: 10.1107/S0907444910007493 | RRID: SCR_014222 | |
| Software, algorithm | PyMOL | Schrödinger | RRID: SCR_000305 | |
| Software, algorithm | Chimera | UCSF | RRID: SCR_004097 | |

## Protein expression and purification

*Xenopus tropicalis* OTOP3 (XtOTOP3, NCBI accession: NP_001106584.1) containing a C-terminal thrombin cleavage site followed by a 10 × His tag was cloned into a pEZT-BM vector (*Morales-Perez et al., 2016*) and heterologously expressed in HEK293F cells (Life Technologies) using the BacMam system (Thermo Fisher Scientific). The baculovirus was generated in Sf9 cells (Life Technologies) following standard protocol and used to infect HEK293F cells at a ratio of 1:40 (virus:HEK293F, v/v) and supplemented with 10 mM sodium butyrate to boost protein expression. Cells were cultured in suspension at 37°C for 48 hr and collected by centrifugation at 3,000 × g. All purification procedures were carried out at 4°C. The cell pellet was re-suspended in buffer A (50 mM Tris, pH 8.0, 200 mM NaCl) supplemented with a protease inhibitor cocktail (containing 2 µg/ml DNase, 0.5 µg/ml pepstatin, 2 µg/ml leupeptin, and 1 µg/ml aprotinin and 0.1 mM PMSF) and homogenized by sonication on ice. XtOTOP3 was extracted with 2% (w/v) n-dodecyl-β-D-maltopyranoside (DDM, Anatrace) supplemented with 0.02% (w/v) cholesteryl hemisuccinate (CHS, Sigma Aldrich) by gentle agitation for 2 hr. After extraction, the supernatant was collected after a 40 min centrifugation at 20,000 rpm and incubated with Ni-NTA resin (Qiagen) by gentle agitation. After 2 hr, the resin was collected on a disposable gravity column (Bio-Rad). The resin was washed with buffer B (50 mM Tris, pH 8.0, 200 mM NaCl, 1 mM DDM and 0.01% (w/v) CHS supplemented with 20 mM imidazole). The washed resin was left on column in buffer B and digested with thrombin (Roche) overnight. After thrombin digestion, the flow-through containing untagged XtOTOP3 was collected, concentrated and purified by size exclusion chromatography on a Superose six column (GE Heathcare) pre-equilibrated with buffer C (20 mM Tris, pH 8.0, 150 mM NaCl, 1 mM DDM and 0.01% (w/v) CHS). The peak fraction was concentrated to 4.7 mg/ml for cryo-EM analysis. HEK293 (CRL-1573) cell lines were purchased from and authenticated by the American Type Culture Collection (ATCC, Manassas, VA). HEK293F cells (RRID: CVCL_D603) were purchased from and authenticated by Thermo Fisher Scientific. The cell lines were not tested for mycoplasma contamination.

## Electron microscopy data acquisition

The cryo-EM grids were prepared by applying 3 µl of XtOTOP3 (4.7 mg/ml) to a glow-discharged Quantifoil R1.2/1.3 300-mesh gold holey carbon grid. Grids were blotted for 4.0 s under 100% humidity at 4°C before being plunged into liquid ethane using a Mark IV Vitrobot (FEI). Micrographs were acquired on a Titan Krios microscope (FEI) operated at 300 kV with a K2 Summit direct electron

detector (Gatan), using a slit width of 20 eV on a GIF-Quantum energy filter. Images were recorded with EPU software (FEI) in super-resolution counting mode with a super resolution pixel size of 0.535 Å. The defocus range was set from −1.5 μm to −3 μm. Each micrograph was dose-fractionated to 30 frames under a dose rate of 4 e-/pixel/s, with a total exposure time of 15 s, resulting in a total dose of about 50 e-/Å2.

## Image processing

Micrographs were motion corrected and binned two times (yielding a pixel size of 1.07 Å per pixel) with MotionCor2 (*Zheng et al., 2017*). The CTF parameters of the micrographs were estimated using the GCTF program (*Zhang, 2016*). All other steps of image processing were performed using RELION2 (*Scheres, 2012*). Initially, ~1000 particles were manually picked from a few micrographs. Class averages representing projections of XtOTOP3 in different orientations were selected from the 2D classification of the manually picked particles, and used as templates for automatic particle picking from the full dataset. 1,276,119 particles were picked from 2761 micrographs. The particles were extracted and binned 3 times (3.21 Å per pixel). After 2D classification, a total of 648,024 particles were finally selected for 3D classification. One of the 3D classes showed good secondary structure features and were selected and re-extracted into the original pixel size of 1.07 Å. After 3D refinement with C2 symmetry imposed, and particle polishing, the resulting 3D reconstructions from ~212,000 particles showed a clear two-fold symmetry with a resolution of 3.9 Å. All resolutions were estimated by applying a soft mask around the protein density and the gold-standard Fourier shell correlation (FSC) = 0.143 criterion. Blocres was used to calculate the local resolution map (*Heymann and Belnap, 2007*).

## Model building, refinement and validation

De novo atomic model building was conducted in Coot (*Emsley et al., 2010*). Amino acid assignment was achieved based mainly on the clearly defined densities for bulky residues (Phe, Trp, Tyr and Arg). Real-space model refinement was performed in Phenix (*Afonine et al., 2018a*), as well as model validation (*Afonine et al., 2018b*). The final structure model of each subunit starts at residue 107 and ends at residue 681. Residues 1–106, 278–302, 342–357, 383–388, 423–505 and 575–605 were disordered and not modeled. Due to poor density quality for TM7 and TM8, residues 357–422 were modeled as poly-alanine. The statistics of the geometries of the models were generated using MolProbity (*Chen et al., 2010*). All the figures were prepared in PyMol (Schrödinger, LLC.) or Chimera (*Pettersen et al., 2004*). Programs used for model building, refinement and validation are compiled by SBGrid. Solvent accessible passages were analyzed using the program CAVER (*Chovancova et al., 2012*).

## Electrophysiology

XtOTOP3, MmOTOP1 and their mutants were cloned into pEGFPC2 vector. About 2 μg of the plasmid containing the N-terminal GFP-tagged XtOTOP3, MmOTOP1 or their mutant was transfected into HEK293 cells grown in a six-well tissue culture dish using Lipofectamine 2000 (Life Technology). Forty-eight hours after transfection, cells were dissociated by trypsin treatment and kept in complete serum-containing medium and re-plated on 35 mm tissue culture dishes in a tissue culture incubator until recording. Similar transfections were performed for the expression test and subcellular localization experiments. In the subcellular localization experiments, 0.5 ug of plasmid containing the plasma membrane marker KRas GTPase tail-tagged with mCherry (*Heo et al., 2006*) was co-transfected with the XtOTOP3.

Patch clamp recordings in the whole-cell and outside-out configurations were used to measure channel activity. The standard intracellular solution contained (in mM): 70 Cesium methanesulfonate (Cs-MS), 2 $MgCl_2$, 1 EGTA, 100 HEPES buffered with Tris, pH = 7.4. The extracellular $Na^+$ free solution contained (in mM): 150 NMDG-MS, 10 HEPES buffered with Tris, pH = 7.4. For pH below 6.5, HEPES was replaced with MES buffered with Tris. For measurement of ion selectivity of the channel, the NMDG-MS in the extracellular solution was replaced with K-MS, Cs-MS or Li-MS respectively.

The data were acquired using an AxoPatch 200B amplifier (Molecular Devices) and a low-pass analogue filter set to 1 kHz. The current signal was sampled at a rate of 20 kHz using a Digidata 1322A digitizer (Molecular Devices) and further analysed with pClamp nine software (Molecular

Devices). Patch pipettes were pulled from borosilicate glass (Harvard Apparatus) and heat-polished to a resistance of 5–10 MΩ filled with the pipette solution. After the patch pipette attached to the cell membrane, a giga-seal (>10 GΩ) was formed by gentle suction. The whole-cell configuration was formed by short zap or suction to rupture the patch, further pulling the pipette away from the cell to form the outside-out configuration. For continuous current recording, the membrane potential was held at −100 mV. To generate the current and voltage relationship, the membrane potential was held at 0 mV, followed by voltage pulses ramp from −100 to +100 or 120 mV over a 150 ms duration.

## Fluorescence Protease Protection (FPP) assay

The FPP assay is adapted from a published protocol (*White et al., 2015*). A Nikon Eclipse Ti fluorescence microscope, equipped with a FITC-3540B-NTE and TRITC-A-NTE filters from Semrock (Rochester, NY, USA) for green and orange fluorescence, respectively, was used for imaging. About 2 μg each of plasmid expressing the N-terminal GFP-tagged wild type XtOTOP3 and pCSCMV:tdTomato plasmid (*Waldner et al., 2009*) expressing free tdTomato fluorescent protein were co-transfected into HEK293 cells grown in a six-well tissue culture dish using Lipofectamine 2000 (Life Technology). Forty-eight hours after transfection, the medium was removed and the cells were washed with KHM buffer (110 mM potassium acetate, 3 mM $MgCl_2$, 20 mM HEPES-NaOH pH7.2) at a flow rate of 5 ml/min for 3 min. The cells were then perfused with KMH buffer containing 50 μg/ml proteinase K at a flow rate of 5 ml/min for 10 min. Images were taken at different time points to monitor the change in green fluorescence. Before cell permeabilization by digitonin, proteinase K was removed by washing the cells three times, 1 min each, with DMEM medium containing 10% FBS, followed by washing the cells with KHM buffer at a flow rate of 5 ml/min for 2 min. 50 μM digitonin in KHM buffer was applied at a flow rate of 5 ml/min for 10 min. Images were taken at different time points to monitor the changes in orange fluorescence. After successful membrane permeabilization indicated by loss of orange fluorescence inside cells, cells were subjected to a second round of proteinase K treatment. Images were taken at different time points to monitor the changes in green fluorescence.

## Acknowledgements

We thank N Nguyen for manuscript preparation. Single particle cryo-EM data were collected at the University of Texas Southwestern Medical Center Cryo-EM Facility that is funded by the CPRIT Core Facility Support Award RP170644. This work was supported in part by the Howard Hughes Medical Institute (YJ) and by grants from the National Institute of Health (GM079179 to Y J) and the Welch Foundation (Grant I-1578 to YJ, grant I-1944-20180324 to XB). XB is supported by the Cancer Prevention and Research Initiative of Texas and Virginia Murchison Linthicum Scholar in Medical Research fund. The authors declare no competing financial interests.

## Additional information

### Funding

| Funder | Grant reference number | Author |
|---|---|---|
| Cancer Prevention and Research Institute of Texas | | Xiao-chen Bai |
| University of Texas Southwestern Medical Center | Virginia Murchison Linthicum Scholar in Medical Research fund | Xiao-chen Bai |
| National Institute of General Medical Sciences | GM079179 | Youxing Jiang |
| Howard Hughes Medical Institute | | Youxing Jiang |
| Welch Foundation | I-1578 | Youxing Jiang |
| Welch Foundation | I-1944-20180324 | Xiao-chen Bai |

The funders had no role in study design, data collection and interpretation, or the decision to submit the work for publication.

### Author contributions

Qingfeng Chen, Conceptualization, Data curation, Formal analysis, Validation, Investigation, Visualization, Writing—original draft, Writing—review and editing, Prepared the samples, Performed data acquisition, image processing and structure determination, Participated in research design, data analysis, and manuscript preparation; Weizhong Zeng, Data curation, Formal analysis, Investigation, Visualization, Writing—review and editing, Performed electrophysiology, Participated in research design, data analysis, and manuscript preparation; Ji She, Data curation, Formal analysis, Investigation, Writing—review and editing, Performed data acquisition, image processing and structure determination, Participated in research design, data analysis, and manuscript preparation; Xiao-chen Bai, Conceptualization, Data curation, Software, Formal analysis, Supervision, Funding acquisition, Validation, Investigation, Methodology, Writing—review and editing, Performed data acquisition, image processing and structure determination, Participated in research design, data analysis, and manuscript preparation; Youxing Jiang, Conceptualization, Resources, Data curation, Supervision, Funding acquisition, Visualization, Project administration, Writing—review and editing, Participated in research design, data analysis, and manuscript preparation

### Author ORCIDs

Ji She (iD) http://orcid.org/0000-0001-7006-6230
Youxing Jiang (iD) http://orcid.org/0000-0002-1874-0504

### Decision letter and Author response

Decision letter https://doi.org/10.7554/eLife.46710.023
Author response https://doi.org/10.7554/eLife.46710.024

# Additional files

### Supplementary files

• Transparent reporting form
DOI: https://doi.org/10.7554/eLife.46710.017

### Data availability

The cryo-EM density map of the XtOTOP3 has been deposited in the Electron Microscopy Data Bank under accession numbers EMDB-0650. Atomic coordinate has been deposited in the Protein Data Bank under accession number 6O84.

The following datasets were generated:

| Author(s) | Year | Dataset title | Dataset URL | Database and Identifier |
|---|---|---|---|---|
| Qingfeng Chen, Xiao-chen Bai, Youxing Jiang | 2019 | Cryo-EM density map of OTOP3 from *xenopus tropicalis* | http://www.ebi.ac.uk/pdbe/entry/emdb/EMD-0650 | Electron Microscopy Data Bank, EMD-0650 |
| Qingfeng Chen, Xiao-chen Bai, Youxing Jiang | 2019 | Atomic coordinates of OTOP3 from *xenopus tropicalis* | http://www.rcsb.org/structure/6O84 | Protein Data Bank, 10.2210/pdb6O84/pdb |

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
