## [Decision Letter]

Thank you for submitting your article "Structural and functional characterization of an otopetrin family proton channel" for consideration by *eLife*. Your article has been reviewed by three peer reviewers, including Baron Chanda as the Reviewing Editor and Reviewer #1, and the evaluation has been overseen by a Reviewing Editor and Richard Aldrich as the Senior Editor. The following individuals involved in review of your submission have agreed to reveal their identity: Sudha Chakrapani (Reviewer #2).

The reviewers have discussed the reviews with one another and the Reviewing Editor has drafted this decision to help you prepare a revised submission.

Summary:

The otopetrin proteins were recently identified as the proton channels involved in perception of sour taste in specialized cells found in the tongue. These receptors were activated by protons and shown to be blocked by μM concentrations of zinc. Here, Chen et al., describe the structure of OTOP3 channel derived from *Xenopus* tropicalis. From a functional standpoint, this channel behaves very much like the mouse orthologs. Interestingly, XtOTOP channel is sensitive to block by both zinc and calcium unlike the mouse channel which is sensitive to zinc block only. This is a very exciting new structure which does not have resemblance to any known protein. The structure shows that the channel has a unique double barrel fold with two possible proton conduction pathways. Based on the structure, the authors identified a conserved triad of residues Gln-Asp/Asn-Tyr. These residues form a structurally conserved constriction point which likely occludes proton permeation. Although not explicitly discussed by the authors, it appears that this structure is of the closed state of the channel. Structure guided mutagenesis shows that residues in the constriction triad are involved in proton permeation and that the two constriction triads do not contribute equally. Overall, this is an excellent structural analysis of a new protein family and is nicely complemented by structure-guided functional analysis. Nevertheless, the reviewers have some concerns mainly with regards to interpretation and presentation of the experimental data. We believe these concerns can be easily addressed in the revised version.

Essential revisions:

1) The authors contend that the rapid decay following activation is both due to inactivation and changes in proton concentration. The reversal potential obtained in a single ramp experiment closely matches the expected E_rev_. Therefore, it is unlikely that the proton concentration changes significantly to cause a rapid current decay following channel opening during a pulse. While the original otopetrin study suggested that the rapid current decrease is due to changes in proton concentration, it seems that the authors data presented here is more consistent with inactivation (or densensitization) mechanism. Have the authors tested the effect on the current decay as the buffer concentration is varied? The D509 is shown to affect current decay and it is concluded that this position is involved in pH-dependent inactivation. Since the experiments were done at one set of pH_o_/pH_i_, this effect could also be explained from drop in conductance (slower accumulation of protons at the intracellular end) from electrostatic effects. Please clarify these points perhaps in the Discussion section.

2) While the mutagenesis data are very nice and show that the conserved residues in the constriction triad are important for proton permeation, I think the interpretation needs further consideration. If the two pores are completely independent, then one would predict that the mutations in one would not have much effect on the currents through the other. In fact, a mutation that completely blocks conduction in one pathway but does not affect the other would just result in reduced current. The data would be indistinguishable from a mutation that just reduces channel expression on the surface. The interesting point here is that some of the single point mutations in C-terminal constriction triad abolish most of the proton current suggesting that either the two pore are tightly linked to each other. The better way to determine the contribution of individual pores is to either express each subdomain separately or block one of the conduction pathways. The alternate explanation is that the permeation pathway is completely different and the two triads are sites for allosteric modulation through a single ion channel pore.

3) Is there similarity or difference from other proton selective channels such as M2 and Hv1? Hydrophobic region consisting of the Gln-Asn (or Asp)-Tyr triad may perhaps have similar function of Phe at the hydrophobic plug of Hv1 and voltage sensor domains of voltage-gated ion channels. Could any titratable residues in the pore (N-pore or C-pore) be predicted as proton acceptor in proton-selective permeation of otopetrin? Related to mutagenesis of N-pore and C-pore, have authors examined if proton selectivity is altered? Some of these points should be addressed in the Discussion section but other can be clarified in your response to the reviewers.

4) What regions contribute to Zn2+ inhibition and what regions may confer differences in Ca^2+^ sensitivity between OTOP1 and OTOP3? As with point #3, we will let you decide whether you want to address these in your response to reviewers or add to the Discussion section.Ca^2+^Discussion section

---

## [Author Response]

Essential revisions:1) The authors contend that the rapid decay following activation is both due to inactivation and changes in proton concentration. The reversal potential obtained in a single ramp experiment closely matches the expected E_rev_. Therefore, it is unlikely that the proton concentration changes significantly to cause a rapid current decay following channel opening during a pulse. While the original otopetrin study suggested that the rapid current decrease is due to changes in proton concentration, it seems that the authors data presented here is more consistent with inactivation (or densensitization) mechanism. Have the authors tested the effect on the current decay as the buffer concentration is varied? The D509 is shown to affect current decay and it is concluded that this position is involved in pH-dependent inactivation. Since the experiments were done at one set of pH_o_/pH_i_, this effect could also be explained from drop in conductance (slower accumulation of protons at the intracellular end) from electrostatic effects. Please clarify these points perhaps in the Discussion section.

Two observations (shown in Figure 2D and E) led us to draw the conclusion that the rapid current decay is due to channel desensitization caused by increase of local intracellular proton concentrations. The first observation is that the channel has almost no activity when intracellular pH is around 6.0. The second observation is that when applying repetitive voltage ramps from -100 mV to 100 mV over a 150 ms duration with a 90 ms interval between two pulses (instead of holding at -100mV in most other experiments), the I-V curves show a progressive decrease of currents accompanied by a progressive shift of the reversal potentials towards 0mV. We intentionally used short pulse duration in the second experiment to reduce the proton influx and thereby slow down the current decay. These points are clarified in the revision. As suggested by the reviewers, we define the current decay of OTOP channel as desensitization instead of inactivation in the revision.

We actually tested two different buffer concentrations in a separate experiment – not discussed in the manuscript – (10 mM and 100 mM) and did not observe any obvious difference in the rate of current decay. It is possible that the proton accumulation occurs locally in a restricted area and cannot be quickly dissipated by the buffer.

When comparing the rate of decay between the WT and D509 mutants, we tried to use cell patches with comparable current levels. The patches containing the wild type channels consistently exhibited a faster current decay regardless of whether the current is small or large. It is unlikely that the slower decay of the D509 mutants is caused by a drop in conductance (slower accumulation of protons at the intracellular end) from electrostatic effects, as the D509E exhibited a similar slower rate of decay as the D509N or D509A mutants. We have included the data for the D509E mutant in the revision. Of course, to absolutely rule out the possibility that channel conductance change in the mutant contribute to the rate of desensitization would require single channel recording, which we are unable to do because the single channel current is too small to measure.

2) While the mutagenesis data are very nice and show that the conserved residues in the constriction triad are important for proton permeation, I think the interpretation needs further consideration. If the two pores are completely independent, then one would predict that the mutations in one would not have much effect on the currents through the other. In fact, a mutation that completely blocks conduction in one pathway but does not affect the other would just result in reduced current. The data would be indistinguishable from a mutation that just reduces channel expression on the surface. The interesting point here is that some of the single point mutations in C-terminal constriction triad abolish most of the proton current suggesting that either the two pore are tightly linked to each other. The better way to determine the contribution of individual pores is to either express each subdomain separately or block one of the conduction pathways. The alternate explanation is that the permeation pathway is completely different and the two triads are sites for allosteric modulation through a single ion channel pore.

We agree that the mutagenesis data at the two triads are a bit confusing and difficult to interpret. Some of the current reduction could be caused by a reduction in protein expression on the cell membrane as suggested by the reviewers. After obtaining the structure of XtOTOP3, we spent more than a year to determine whether one or both pores are ion conductive, including expressing the N-/C-pore separately and using MTS reagent to block the pore, as suggested by reviewers. We were unable to get any currents when separately expressing the N- or C-pore constructs, which might be caused by compromised protein expression, folding or trafficking. The MTS reagent blockage experiments were unsuccessful also, since the WT channel contains multiple cysteines that could already be blocked by the MTS reagent from the extracellular side. Furthermore, mutations of these cysteines resulted in the loss of channel activity. The only conclusion we can draw based on the current data is that both triads appear to be sensitive to mutations and mutations at the C-pore triad appears to have a stronger effect. We also agree with the reviewers that we cannot rule out the alternative possibility that the permeation pathway is completely different and that the two triads are sites for allosteric modulation through a single ion channel pore. In the revision, we briefly raised this possibility in the Discussion section.

3) Is there similarity or difference from other proton selective channels such as M2 and Hv1? Hydrophobic region consisting of the Gln-Asn (or Asp)-Tyr triad may perhaps have similar function of Phe at the hydrophobic plug of Hv1 and voltage sensor domains of voltage-gated ion channels. Could any titratable residues in the pore (N-pore or C-pore) be predicted as proton acceptor in proton-selective permeation of otopetrin? Related to mutagenesis of N-pore and C-pore, have authors examined if proton selectivity is altered? Some of these points should be addressed in the Discussion section but other can be clarified in your response to the reviewers.

The OTOP channel is very different from M2 channels, which form a tetramer containing a histidine tetrad that defines the channel’s proton selectivity and gating mechanism. The triad in the OTOP channel is also quite different from the hydrophobic plug (or gating charge transfer center formed by two acidic residues and an aromatic residue) of the Hv1 and VSD of voltage-gated channels. In the Hv1 channel, the aspartate is essential for proton selectivity and its mutation to neutral residues renders the channel anion permeable. In most OTOP channels, the triad has the sequence Gln-Asn-Tyr and does not contain titratable Glu or Asp. We measured the selectivity of several triad mutants (Q232E, D262Q, Y322W, N623Q and Y666A) that retain some channel activity and found no obvious change in proton selectivity. We should point out that it is still unclear whether the triads or some other regions define the proton selectivity in OTOP channels. We feel it is premature to discuss the selectivity mechanism with the assumption that the constriction triads are directly responsible for proton selectivity.

4) What regions contribute to Zn2+ inhibition and what regions may confer differences in Ca^2+^ sensitivity between OTOP1 and OTOP3? As with point #3, we will let you decide whether you want to address these in your response to reviewers or add to the Discussion section.

We do not know what regions contribute to Zn^2+^ inhibition in OTOP channels and what regions may confer differences in Ca^2+^ sensitivity between OTOP1 and OTOP3. We have spent quite some efforts on mutagenesis studies on the negatively charged residues on the extracellular side, but have not yet been able to pinpoint the Zn^2+^ and Ca^2+^ inhibition sites. Efforts aimed at solving the XtOTOP3 structures in complex with Zn^2+^ or Ca^2+^ were not successful due to the poor resolution of the reconstructed maps. In addition, the structural mechanism of proton-dependent activation of the channel is also unclear. All these points raised by reviewers are important questions remained to be addressed in our ongoing studies. However, due to a lack of any experimental data at the current stage, we are not in a position to provide any speculative discussion on these issues.